# The association between paternal characteristics and exclusive breastfeeding in Ghana

**Frank Kyei-Arthur**[ID]**¤\*◉, Martin Wiredu Agyekum◉, Grace Frempong Afrifa-Anane¤◉**

Regional Institute for Population Studies, University of Ghana, Legon, Accra, Ghana

◉ These authors contributed equally to this work.
¤ Current address: Department of Environment and Public Health, University of Environment and Sustainable Development, Somanya, Eastern Region, Ghana
\* fkyei-arthur@uesd.edu.gh

**Data Availability Statement:** All relevant data are within the manuscript and its Supporting Information files.

**Funding:** The authors received no specific funding for this work.

## Abstract

### Background

Studies have shown that partners play an influential role in exclusive breastfeeding practice and that they can act as either deterrents or supporters to breastfeeding. However, there are limited studies on the influence of partners' characteristics on exclusive breastfeeding in Ghana. This study examined the association between partners' characteristics and exclusive breastfeeding in Ghana.

### Methods

This cross-sectional study used data from the 2014 Ghana Demographic and Health Survey. Infants less than 6 months old (exclusively breastfed or not) with maternal and paternal characteristics were included in the study. A total of 180 participants were used for the study. A binary logistic regression was used to examine the influence of partners' characteristics on exclusive breastfeeding.

### Results

Partners' characteristics such as education, desire for children, religion, and children ever born were associated with exclusive breastfeeding. Mothers whose partners had primary education (AOR = 0.12; CI 95%: 0.02–0.93; p = 0.04) were less likely to practice exclusive breastfeeding compared to those whose partners had no formal education. Also, mothers whose partners desired more children (AOR = 0.20; CI 95%: 0.06–0.70; p = 0.01) were less likely to practice exclusive breastfeeding compared to those whose partners desire fewer children.

### Conclusion

Improving EBF requires the involvement of partners in exclusive breastfeeding campaigns/ programmes. A more couple-oriented approach is required by health practitioners to

**Competing interests:** The authors have declared that no competing interests exist.

educate and counsel both mothers and partners on the importance of exclusive breastfeeding in Ghana.

## Introduction

Exclusive breastfeeding (EBF) has been identified as an essential practice to enhance the nutritional status and growth of infants aged under 6 months [1]. EBF is the practice of giving infants only breast-milk and no liquids or solids with the exceptions of drops or syrups consisting of vitamins, mineral supplements, or medicines [2]. The World Health Organisation (WHO) and United Nations Children's Fund (UNICEF) recommend EBF for the first six months of an infant's life since breastmilk contains all the essential nutrients required by an infant to attain optimal growth and health [1]. Exclusive breastfeeding has enormous benefits for mothers and their infants. It reduces neonatal and infant mortality and morbidity, enables infants to crawl early, enhances growth and cognitive development, and reduces the risk of childhood obesity [3–7]. In addition, it helps mothers to lose weight after birth, reduces their risk of pregnancy in the first 6 months after birth, and reduces their risk of breast and ovarian cancers [3, 8, 9]. It is estimated that increasing breastfeeding could annually help avert 823,000 child deaths and 20,000 breast cancer deaths [10].

Despite these benefits, the practice of EBF is generally low [11]. In 2019, two-fifth (41%) of infants aged under 6 months were exclusively breastfed worldwide. The WHO estimates that at least 50% and 70% of infants aged under 6 months globally should be exclusively breastfed by 2025 and 2030 respectively [12, 13]. Ghana's EBF rate rose steadily from 53% in 2003 to 63% in 2008. The increase in Ghana's EBF rate can be attributed to the implementation of policies and programmes such as the Breastfeeding Promotion Regulations 2000 [14], Imagine Ghana Free of Malnutrition [15], and Infant and Young Child Feeding Strategy [16]. These policies and programmes highlighted the importance of practicing exclusive breastfeeding. For instance, the Breastfeeding Promotion Regulations 2000 prohibits the sale and promotion of infant formula or any other product marketed as appropriate for feeding infants less than six months of age in health facilities or public places [14]. It is worth noting that these policies and programmes placed less emphasis on couple-oriented approaches to enhance exclusive breastfeeding.

However, a total of 52% of infants below 6 months were breastfed exclusively in 2014 [17]. This is a reduction upon the 2008 rate of 63%, and thus, calls for further investigation. The decline in exclusive breastfeeding can be attributed to factors such as violation of the Ghana Breastfeeding Promotion Regulation, selective implementation of funded breastfeeding initiatives at the district and regional levels, and increased participation of women in the labour market [18, 19]. Participation of women in the labour market, especially formal employment has been linked with a decline in the practice of exclusive breastfeeding since their workplace and working conditions do not support the practice of exclusive breastfeeding [20, 21].

Previous studies in Ghana have established that mothers breastfeeding practices are influenced by interpersonal and community factors including pressure from family members and cultural beliefs and practices [22, 23]. For instance, Diji et al.'s [24] study among mothers at a public health facility in Kumasi South Hospital revealed that the belief that breastmilk is inadequate to meet the nutritional needs of infants and pressure from family members to give water and food to infant hindered the practice of exclusive breastfeeding. Studies have also documented the traditional practice of giving herbal concoctions to infants to drink after birth, especially in the Northern part of Ghana. It is perceived that herbal concoctions make the infants strong and healthy [25–27].

Evidence suggests that a partner (a man who is either currently married or living with a woman) is one of the most influential persons to the mother and that he can act as either a deterrent or supporter to breastfeeding [28–31]. For instance, a father can provide the practical and emotional support needed for successful breastfeeding, contribute to maternal breastfeeding confidence, and influence decisions on the duration of breastfeeding [32, 33]. In addition, studies in Brazil and Sweden have found that paternal age and education influence the practice of exclusive breastfeeding [34, 35].

However, studies on EBF in sub-Saharan Africa, particularly Ghana, have mainly focused on maternal characteristics, child characteristics, household characteristic (such as household size), family influence (grandmothers and in-laws), health-related factors (such as place of delivery), and spatial characteristics (such as place of work) [20, 24, 36–39]. There is a dearth of information on the influence of partners on EBF. In light of the limited literature and reduction in exclusive breastfeeding rate in Ghana, this study seeks to examine the influence of partners' characteristics on EBF in Ghana using a nationally representative sample. Understanding the influence of partners' characteristics on EBF in addition to other factors will provide a holistic understanding to help develop appropriate interventions to promote exclusive breastfeeding among mothers in Ghana.

## Materials and methods

### Data and study population

This study used data from the 2014 Ghana Demographic and Health Survey (GDHS). The GDHS is a nationally representative sample survey that is conducted every five years and the 2014 GDHS is the sixth round in the series. A two-stage sample design was employed to select participants for the interview. The 2014 GDHS sampling procedure and methodology have been described in detail in the Ghana Demographic and Health Survey manual [18]. The 2014 Ghana Demographic and Health Survey had a sample of 5,884 infants. However, the analysis of the current study was based on a sample size of 180 infants less than six months (exclusively breastfed or not) who had both maternal and paternal characteristics.

### Variables

**Dependent variable.** The dependent variable for the study was exclusive breastfeeding. Exclusive breastfeeding was defined based on the World Health Organisation 24-hour recall method. This was measured as a binary variable using several food items such as breastmilk, water, liquids, milk, and solid food. Infants who were fed only on breast milk 24 hours preceding the survey were coded as "1" for exclusive breastfeeding while infants who were fed on breastmilk and any other food 24 hours preceding the survey were coded as "0" and classified as non-exclusive breastfeeding.

**Independent variables.** Partners' characteristics such as age, education, occupation, religion, ethnicity, place of residence, partners' attitude towards wife beating, desire for children, and number of children by partner were the independent variables for the study. These variables were selected based on literature review. Partners' age was categorized as 15–24 years, 25–34 years, and 35 years and above. The level of education of a partner was re-categorized as no education, primary, and secondary, and above. The proportion of partners with higher education was few so they were added to partners who had secondary education. Partners' main occupation was categorized as professionals, sales and service, agriculture, skilled manual, and unskilled manual. In addition, partners' religion was classified as Christians, Muslims, and Other comprising Traditionalist, Buddhist, and Hinduism. Partners' ethnicity was categorized as Ewe, Ga/Dangme, Akan, Mole-Dagbani, Gurma, and other including Grussi and Mande.

Partner's place of residence was categorized as urban and rural areas while attitude towards wife beating was classified as favourable and unfavourable. Moreover, a partner's desire for having children was classified as a partner desiring for fewer children than the mother, a partner desiring for more children than the mother, both the partner desire for the same number of children. Lastly, children ever born was measured as the total number of children born by the partner.

**Control variables.**   The study controlled for other variables that may influence exclusive breastfeeding based on the review of the literature. The control variables were mother's age (15–24, 25–34, and 35 years and above), mother's education (no education, primary, and secondary and above), mother's marital status (married and living with a partner), mother's children ever born (single child mothers and multiparous mothers), decision on health (woman alone, partner alone, and both partner and woman), mother's ethnicity (Akan, Ewe, Mole-Dagbani, Gurma and other groups comprising Ga/Dangme, Guan, and Grusi), frequency of antenatal clinic, maternity leave (No, and Yes), postnatal (Yes and No), mother's attitude towards wife beating (favourable and unfavourable), sex of the child (male and female), age of infant (less than one month, 1 month, 2 months, 3 months, 4 months and 5 months) and birth order. All variables and their categories used in this study align with the standards for Ghana.

**Data analysis.**   The data was analyzed using Stata version 15. The data were weighted to make it representative and to provide a better statistical estimate. The proportion of each variable was described using tables and percentages. A binary logistic regression model was used to examine the relationship between partners' characteristics, and exclusive breastfeeding, controlling for other variables. A binary logistic regression was used because the dependent variable (exclusive breastfeeding) is a dichotomous variable "with responses "yes" and "no". All associations were tested at a significance level of 0.05.

## Results

### Characteristics of partners

Table 1 shows the exclusive breastfeeding practice and partner's characteristics. More than half (58.33%) of the infants were exclusively breastfed 24 hours preceding the survey. About six out of ten (58.33%) partners were aged 35 years and older, and about 58% of the partners had at least secondary or higher education. Also, about 32% of partners had no formal education. The highest proportion (48.9%) of partners were engaged in agricultural work and a few (8.3%) were into sales and services. In addition, more than half (53.89%) of the partners were Christians and about three out of ten (31.11%) partners belonged to the Mole-Dagbani ethnic group. Most partners were living in rural areas (56.67%) and had favourable attitude towards wife beating (82.78%). With regards to partner desire for children, about 46% of both mothers and their partners wanted the same number of children. Additional socio-demographic characteristics of the partners are presented in Table 1.

### Characteristics of mothers and infants

The results indicate that more than two-fifths (43.89%) of the mothers were within the age group 25–34 years and the highest proportion (40.6%) of the mothers had no formal education (Table 2). Also, seven out of ten mothers (76.11%) were married and the majority (89.44%) of the mothers were multiparous mothers. Regarding decision on health, more than half (56.11%) of the mothers made a joint decision with their partner. Furthermore, the majority (86.7%) of the mothers had no maternity leave and 68.89% of them attended postnatal clinics. Regarding the infants, more than half (53.89%) were males and the highest proportion

**Table 1. Description of exclusive breastfeeding and partner's characteristics.**

| Variables | Frequency | Percentage |
|---|---|---|
| **Exclusive breastfeeding** | | |
| No | 75 | 41.67 |
| Yes | 105 | 58.33 |
| **Partner's age** | | |
| 15–24 | 9 | 5.00 |
| 25–34 | 66 | 36.67 |
| 35+ | 105 | 58.33 |
| **Partner's educational level** | | |
| No education | 57 | 31.67 |
| Primary | 18 | 10.00 |
| Secondary/higher | 105 | 58.33 |
| **Partner's occupation** | | |
| Professionals | 22 | 12.22 |
| Sales and service | 15 | 8.33 |
| Agriculture | 88 | 48.89 |
| skilled manual | 30 | 16.67 |
| Unskilled manual | 25 | 13.89 |
| **Partner's religion** | | |
| Christian | 97 | 53.89 |
| Muslims | 45 | 25.00 |
| Other | 38 | 21.11 |
| **Partner's ethnicity** | | |
| Ewe | 22 | 12.22 |
| Ga/Dangme | 11 | 6.11 |
| Akan | 46 | 25.56 |
| Mole-Dagbani | 56 | 31.11 |
| Gurma | 26 | 14.44 |
| Other | 19 | 10.56 |
| **Partner's place of residence** | | |
| Urban | 78 | 43.33 |
| Rural | 102 | 56.67 |
| **Partner's attitude towards wife beating** | | |
| Unfavourable | 149 | 82.78 |
| Favourable | 31 | 17.22 |
| **Partner's desire for children** | | |
| Partner wants fewer | 47 | 26.11 |
| Partner wants more | 51 | 28.33 |
| Both want same | 82 | 45.56 |
| | **Min, Max** | **Mean, std** |
| **Partner's children ever born** | 1, 21 | 4.71, 3.60 |

(22.78%) of the infants were aged five months old. Additional socio-demographic characteristics of the mothers and infants are presented in Table 2.

## Predictors of exclusive breastfeeding

Table 3 illustrates factors associated with exclusive breastfeeding in Ghana. The model shows the effects of partner, mother, and infant characteristics on exclusive breastfeeding. The

**Table 2. Description of mother and infant characteristics.**

| Variables | Frequency | Percentage |
|---|---|---|
| **Age of mother** | | |
| 15–24 | 47 | 26.11 |
| 25–34 | 79 | 43.89 |
| 35+ | 54 | 30.00 |
| **Education of mother** | | |
| No education | 73 | 40.56 |
| Primary | 40 | 22.22 |
| Secondary and higher | 67 | 37.22 |
| **Marital status of mother** | | |
| Married | 137 | 76.11 |
| Living with partner | 43 | 23.89 |
| **Mother's children ever born** | | |
| Single | 19 | 10.56 |
| Multiparous | 161 | 89.44 |
| **Decision on health** | | |
| Respondent alone | 22 | 12.22 |
| Respondent and partner | 101 | 56.11 |
| Partner alone | 57 | 31.67 |
| **Mother's ethnicity** | | |
| Ewe | 23 | 12.78 |
| Akan | 43 | 23.89 |
| Mole Dagbani | 66 | 36.67 |
| Other | 48 | 26.67 |
| **Maternity leave** | | |
| No | 156 | 86.67 |
| Yes | 24 | 13.33 |
| **Postnatal** | | |
| No | 56 | 31.11 |
| Yes | 124 | 68.89 |
| **Mother's attitude towards wife beating** | | |
| Unfavourable | 116 | 64.44 |
| Favourable | 64 | 35.56 |
| **Sex of infant** | | |
| Male | 97 | 53.89 |
| Female | 83 | 46.11 |
| **Age of infant** | | |
| 0 | 13 | 7.22 |
| 1 | 33 | 18.33 |
| 2 | 36 | 20.00 |
| 3 | 27 | 15.00 |
| 4 | 30 | 16.67 |
| 5 | 41 | 22.78 |
| | **Min, Max** | **Mean, std** |
| **Partner's children ever born** | 1, 21 | 4.71, 3.60 |
| **Frequency of ANC** | 0, 98 | 7.14, 10.13 |
| **Birth order** | 1, 10 | 3.74, 2.10 |

**Table 3. Binary logistic regression showing the factors associated with exclusive breastfeeding.**

| Variables | Exclusive breastfeeding (EBF) | | |
|---|---|---|---|
| | Adjusted Odds Ratio (AOR) | 95% CI | P-value |
| **Partner's educational level** | | | |
| No education (RC) | | | |
| Primary | 0.12 | 0.02–0.93 | **0.04** |
| Secondary/higher | 0.50 | 0.10–2.55 | 0.40 |
| **Partner's occupation** | | | |
| Professionals (RC) | | | |
| Sales and service | 0.38 | 0.05–3.05 | 0.36 |
| Agriculture | 3.52 | 0.51–24.49 | 0.20 |
| Skilled manual | 2.53 | 0.49–13.09 | 0.27 |
| Unskilled manual | 0.48 | 0.10–2.43 | 0.38 |
| **Partner's age** | | | |
| 15–24 (RC) | | | |
| 25–34 | 1.09 | 0.12–9.97 | 0.94 |
| 35+ | 0.97 | 0.09–10.63 | 0.98 |
| **Partner's desire for children** | | | |
| Partner wants fewer (RC) | | | |
| Partner wants more | 0.20 | 0.06–0.70 | **0.01** |
| Both wants same | 0.33 | 0.10–1.13 | 0.08 |
| **Partner's religion** | | | |
| Christian (RC) | | | |
| Muslims | 0.20 | 0.05–0.76 | **0.02** |
| Other | 2.59 | 0.64–10.56 | 0.18 |
| **Partner's ethnicity** | | | |
| Ewe (RC) | | | |
| Ga/Dangme | 2.17 | 0.22–21.96 | 0.51 |
| Akan | 1.29 | 0.18–9.40 | 0.80 |
| Mole-Dagbani | 2.36 | 0.15–37.74 | 0.55 |
| Gurma | 0.39 | 0.02–6.61 | 0.51 |
| Other | 4.12 | 0.32–53.06 | 0.28 |
| **Partner's place of residence** | | | |
| Urban (RC) | | | |
| Rural | 0.75 | 0.19–3.03 | 0.69 |
| **Partner's children ever born** | 1.38 | 1.08–1.77 | **0.01** |
| **Partner's attitude towards wife beating** | | | |
| Unfavourable (RC) | | | |
| Favourable | 0.44 | 0.14–1.39 | 0.16 |
| **Age of mother** | | | |
| 15–24 (RC) | | | |
| 25–34 | 0.97 | 0.27–3.48 | 0.97 |
| 35+ | 0.20 | 0.03–1.21 | 0.08 |
| **Education of mother** | | | |
| No education (RC) | | | |
| Primary | 1.79 | 0.39–8.18 | 0.45 |
| Secondary and higher | 5.14 | 0.95–27.91 | 0.06 |
| **Marital status of mother** | | | |
| Married (RC) | | | |

(*Continued*)

**Table 3.** (Continued)

| Variables | Exclusive breastfeeding (EBF) | | |
|---|---|---|---|
| | Adjusted Odds Ratio (AOR) | 95% CI | P-value |
| Living with partner | 1.49 | 0.45–4.93 | 0.51 |
| **Mother's ethnicity** | | | |
| Ewe (RC) | | | |
| Akan | 0.15 | 0.02–1.10 | 0.06 |
| Mole Dagbani | 0.50 | 0.04–6.65 | 0.60 |
| Other | 0.55 | 0.06–5.14 | 0.60 |
| **Mother's children ever born** | | | |
| Single (RC) | | | |
| Multiparous | 1.30 | 0.23–7.17 | 0.77 |
| **Decision on health** | | | |
| Respondent alone (RC) | | | |
| Respondent and partner | 2.63 | 0.63–10.96 | 0.18 |
| Partner alone | 0.99 | 0.21–4.63 | 0.99 |
| **Frequency of antenatal visits** | 1.08 | 1.00–1.16 | **0.04** |
| **Maternity leave** | | | |
| No (RC) | | | |
| Yes | 2.02 | 0.43–9.43 | 0.37 |
| **Postnatal** | | | |
| No (RC) | | | |
| Yes | 0.71 | 0.24–2.09 | 0.54 |
| **Mother's attitude towards wife beating** | | | |
| No (RC) | | | |
| Yes | 1.14 | 0.36–3.65 | 0.82 |
| **Sex of child** | | | |
| Male (RC) | | | |
| Female | 0.69 | 0.28–1.71 | 0.42 |
| **Age of child** | | | |
| Less than 1 (RC) | | | |
| 1 | 0.07 | 0.00–1.19 | 0.07 |
| 2 | 0.03 | 0.00–0.41 | **0.01** |
| 3 | 0.01 | 0.00–0.23 | **0.00** |
| 4 | 0.01 | 0.00–0.18 | **0.00** |
| 5 | 0.01 | 0.00–0.12 | **0.00** |
| Birth order | 0.95 | 0.64–1.42 | 0.82 |

RC: Reference category; R square = 0.3460.

R-square of the model (0.3460) shows that about 34.60% of the partner, maternal and infant characteristics explain variations in EBF practice in Ghana.

The results of the adjusted regression model indicate that partner's education, partner's desire for children, partner's religion, and partner's children ever born were significantly associated with EBF.

Mothers whose partners had primary education were less likely to practice EBF compared to those whose partners had no formal education (AOR = 0.12; CI 95%: 0.02–0.93; p = 0.04). Also, mothers whose partners desire more children were less likely to practice EBF compared to those whose partners desire fewer children (AOR = 0.20; CI 95%: 0.06–0.70; p = 0.01). In

addition, mothers whose partners were Muslims were less likely to practice EBF compared to mothers whose partners were Christians (AOR = 0.20; CI 95%: 0.05–0.76; p = 0.02). Moreover, as the number of children by a partner increases, the odds of a mother practicing EBF increases (AOR = 1.38; CI 95%: 1.08–1.77; p = 0.01).

Other factors that were found to predict EBF include frequency of antenatal visits and age of infant. Regarding frequency of antenatal care visits, an increase in antenatal care visits increases the practice of exclusive breastfeeding by mothers (AOR = 1.08; CI 95%: 1.00–1.16; p = 0.04).

Lastly, infants who were two months old were less likely to be exclusively breastfed compared to infants less than one month old (AOR = 0.03; CI 95%: 0.00–0.41; p = 0.01). Also, infants who were three months old were less likely to be exclusively breastfed compared to infants less than one year old (AOR = 0.01; CI 95%: 0.00–0.23; p = 0.00). Furthermore, infants who were four months old were less likely to be exclusively breastfed compared to infants less than one year old (AOR = 0.01; CI 95%: 0.00–0.18; p = 0.000).

## Discussion

Considering the reduction of EBF practice in Ghana, identifying partner characteristics that influence EBF would help develop policies and other interventions to increase EBF prevalence in Ghana. In the light of scant literature on significant role partners' play in EBF, we examined the influence of partner's characteristics on exclusive breastfeeding (EBF) in Ghana. Our findings unearth the relevance of partner's education, desire for children, religion, and children ever born in the practice of EBF which have implications for policy and research. In addition, the results of the study show that mother's antenatal visits and age of child explain mothers' exclusive breastfeeding behaviour in Ghana.

We found that EBF was practiced by more than half (58.33%) of mothers in this study. The results indicated that more than half of the mothers practiced EBF in Ghana. However, the proportion of mothers practicing EBF is below the recommended rate of 90% by WHO [40]. This, therefore, has implications for the welfare of children in relation to morbidity and mortality. Comparing the prevalence of EBF (58.33%) in this study to the 2008 Ghana Demographic and Health Survey EBF prevalence (63.00%), there is a decline in the practice of EBF. Despite the decline in the practice of EBF in Ghana, other child nutrition indicators (such as stunting, wasting, and underweight) have improved over time. Since 1988, the prevalence of stunting, wasting, and underweight has reduced from about 34%, 9%, and 23% to 19%, 5%, and 11% respectively [17].

The reduction of EBF practice could therefore explain the slight decline of neonatal mortality from 43 per 1000 live births in 2003 to 29 per 1000 live births in 2014 [17]. There is a need for measures and policies to be strengthened to promote the practice of EBF. For instance, Tsai [41] reported that encouragement of mothers to use lactation rooms and milk expression breaks programme increased breastfeeding in Taiwan. Similarly, the involvement of men in breastfeeding programmes in Brazil led to an increase in exclusive breastfeeding practice [42]. In addition, assistance with preventing and managing lactation difficulties among couples in Italy improved exclusive breastfeeding [43].

Although formal education has been recognized to have a positive effect on attitude and health-related behaviours [44, 45], findings from the present study indicate otherwise. The likelihood to practice EBF was lower among mothers whose partners had primary education compared to those whose partners had no formal education. The finding is consistent with previous studies in Sweden and India which found that partners with low education are less likely to practice exclusive breastfeeding compared to those with no education [34, 46]. A

probable reason is that having knowledge about behaviour does not always translate into an attitude or behavioural change [47]. Therefore, mothers whose partners have acquired some level of formal education may not necessarily be empowered to practice EBF.

Furthermore, the findings revealed that the likelihood to practice EBF was lower among mothers whose partners desired more children than those whose partners desired fewer children. The practice of EBF reduces mothers' risk of pregnancy in the first 6 months after birth. This is because it serves as a family planning method and could delay mothers in conceiving and having children [3, 9].

It was also evident in the findings that the likelihood to exclusively breastfeed was lower among mothers whose partners were Muslims compared to those whose partners were Christians. This finding is consistent with previous studies [23]. The Holy Quran encourages breastfeeding among Muslims [48]. However, the reasons why mothers whose partners were Muslims are less likely to exclusively breastfeed compared to those whose partners are Christians could not be explained in the present study and therefore further studies using qualitative approaches could explore the probable reasons.

The findings also indicated a positive association between partner's number of children ever born and practice of EBF. The reasons underlying this finding could not be explained in the present study due to data limitation and therefore needs further exploration using qualitative approaches.

Importantly, partner characteristics such as age, occupation, place of residence, and attitude towards wife beating, were not significantly associated with the practice of EBF. This finding is similar to other studies in Malaysia [49] and Brazil [50] which found that paternal age is not significantly associated with exclusive breastfeeding. Despite the non-significant association between occupation and EBF, mothers whose partners were engaged in agriculture were more likely to practice EBF than those whose partners were professionals. A plausible explanation is that in Ghana, agricultural extension officers, (for example, Women in Agricultural Development (WIAD) officers) provide health and nutrition education to farmers during the provision of extension services to enhance the wellbeing of their households, including the practice of exclusive breastfeeding [51].

Furthermore, there was a positive association between antenatal care visits and the practice of EBF. Previous studies have established that antenatal clinics provide an opportunity for health care workers (midwives, community health nurses) to interact and educate mothers on infant feeding, the nutritional value of EBF, and challenges associated with EBF. The health education provided by health care workers during antenatal care visits increases knowledge on infant feeding and thus, enhances the practice of exclusive breastfeeding [52–54].

Finally, the findings showed that the likelihood to exclusively breastfeed decreased as the age of an infant increased. Similar findings have been reported in Nigeria [55, 56] and India [57]. It is explained that most mothers perceive that as an infant's age increases, he/she does not get satisfied when exclusively breastfed. This leads to persistent crying and sleeplessness. The introduction of complementary foods, therefore, reduces hunger and calms the infants [58]. In addition, Ghanaian women working in the formal sector are entitled to three-month maternity leave after delivery. Therefore, the early return to work tends to affect exclusive breastfeeding especially when the infant is aged three months or above [59]. The introduction of complementary foods as infants get older enables the mothers to attend to other activities particularly work [56, 60]. This calls for the need for the Ministry of Health and other government agencies to advocate for a longer duration of maternity leave since exclusive breastfeeding is beneficial to the mother and infant.

Results of this study provide an understanding of the influence of partners in the practice of EBF. This study is very relevant towards the inclusion of partners in exclusive breastfeeding

campaigns. Countries (such as Brazil and Taiwan) that have included men in breastfeeding-specific interventions have succeeded in improving exclusive breastfeeding practice [42, 43]. In Ghana, there are no direct policies or measures targeting partners in breastfeeding practices. Programmes involving partners and women such as partners offering assistance with preventing and managing lactation difficulties, partners helping with household tasks and child care, partners encouraging their women to use lactation room and milk expression breaks could help improve exclusive breastfeeding practice in Ghana. Findings from this study could inform policy makers to encourage men to follow women to antenatal clinics. Through this, they will be educated on the importance of exclusive breastfeeding and on how to manage the challenges associated with it. In addition, there could be breastfeeding schools for partners to help them understand breastfeeding, other maternal and child issues as well as addressing concerns of partners related to exclusive breastfeeding.

## Limitations

This study had some limitations that are worth noting. First, due to the cross-sectional nature of the study, we cannot infer causality. Second, the sample size for the study was small (n = 180) because some data on partners of mothers with an infant less than six months were missing. Third, exclusive breastfeeding was limited to 24 hours preceding the survey. This does not accurately measure the exclusive breastfeeding of infants less than six months. Despite these limitations, this study is one of the few to examine the influence of partners' characteristics on exclusive breastfeeding. The findings, therefore, contribute to exclusive breastfeeding literature in sub-Saharan Africa.

## Conclusions

Findings from this study indicate an association between partners' characteristics and EBF. As found in this study, partner's education, religion, desire for more children, and children ever born significantly predicted the practice of EBF. This is an indication that partners play a crucial role in the practice of EBF in Ghana. The Global Strategy for Infant and Young Child Feeding emphasizes the need for those involved in promoting breastfeeding to understand the benefits and importance of exclusive breastfeeding. Therefore, health policy makers should provide incentives to partners to promote their involvement in attaining optimal nutrition and breastfeeding. To achieve the full benefit of partners' involvement, future research should encompass qualitative approaches to help understand and explain the association between the partners' characteristics and exclusive breastfeeding.

## Supporting information

**S1 Data.**
(XLSX)

**S2 Data.**
(XLSX)

## Acknowledgments

The authors acknowledge the Measure Demographic and Health Survey programme for making the 2014 Ghana Demographic and Health Survey data set available for this study. The authors are grateful to the academic editor and reviewers for their constructive comments and suggestions.

## Author Contributions

**Conceptualization:** Frank Kyei-Arthur, Grace Frempong Afrifa-Anane.

**Formal analysis:** Martin Wiredu Agyekum.

**Supervision:** Frank Kyei-Arthur, Martin Wiredu Agyekum, Grace Frempong Afrifa-Anane.

**Validation:** Frank Kyei-Arthur, Martin Wiredu Agyekum, Grace Frempong Afrifa-Anane.

**Writing – original draft:** Frank Kyei-Arthur, Martin Wiredu Agyekum, Grace Frempong Afrifa-Anane.

**Writing – review & editing:** Frank Kyei-Arthur, Martin Wiredu Agyekum, Grace Frempong Afrifa-Anane.

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
