## [Decision Letter · Decision Letter 0]

10 Mar 2021

PONE-D-20-36508

Partner characteristics and exclusive breastfeeding in Ghana

PLOS ONE

Dear Dr. Kyei-Arthur,

Thank you for submitting your manuscript to PLOS ONE. After careful consideration, we feel that it has merit but does not fully meet PLOS ONE’s publication criteria as it currently stands. Therefore, we invite you to submit a revised version of the manuscript that addresses the points raised during the review process.

Paternal influence on exclusive breastfeeding practices are an important consideration, although a significant gap in our knowledge exists - especially from your region. Following the suggestions of the two peer-reviews will certainly strengthen, focus and clarify a number of aspects of your work. 

We look forward to receiving your revised manuscript.

Kind regards,

Joann M. McDermid, MSc, PhD, RDN, FAND

Academic Editor

PLOS ONE

Journal Requirements:

2.We note that you have indicated that data from this study are available upon request. PLOS only allows data to be available upon request if there are legal or ethical restrictions on sharing data publicly. For more information on unacceptable data access restrictions, please see http://journals.plos.org/plosone/s/data-availability#loc-unacceptable-data-access-restrictions.

3. For studies involving humans categorized by race/ethnicity, age, disease/disabilities, religion, sex/gender, sexual orientation, or other socially constructed groupings, authors should: 1) Explicitly describe their methods of categorizing human populations, 2) Define categories in as much detail as the study protocol allows, 3) Justify their choices of definitions and categories, 4) Explain whether (and if so, how) they controlled for confounding variables such as socioeconomic status, nutrition, environmental exposures, or similar factors in their analysis.

Reviewers' comments:

Reviewer's Responses to Questions

**Comments to the Author**

1. Is the manuscript technically sound, and do the data support the conclusions?

Reviewer #1: Partly

Reviewer #2: Partly

2. Has the statistical analysis been performed appropriately and rigorously? 

Reviewer #1: No

Reviewer #2: Yes

3. Have the authors made all data underlying the findings in their manuscript fully available?

Reviewer #1: Yes

Reviewer #2: Yes

4. Is the manuscript presented in an intelligible fashion and written in standard English?

Reviewer #1: No

Reviewer #2: Yes

5. Review Comments to the Author

Reviewer #1: This is a very important topic that the researchers have decided to study. Overall, they did a good job approaching this study. However, there are some major revisions necessary before considering for review for resubmission.

Consider making the title more descriptive.

Rationale/justification for the study needs to be stronger. What specifically is this adding to the current literature?

If the analyses were adjusted, that needs to be explained and adjusted results should be listed as AORs instead of ORs.

Description of inclusion criteria should be more specific in the abstract methods and methods section.

Results should be presented with 2 decimals; remove "95%" from methods

Conclusion in abstract should be more specific. How do these findings help? How can they inform policies and interventions? What should public health professionals do differently in Ghana based on your findings?

Are all authors are the same institution? If yes, that should be clarified on the title page.

Line 18 should read sub-Saharan

Line 43 Odd to reference EBF as an intervention; consider calling it an "essential practice"

Be sure to be consistent using EBF throughout - see Line 44

Line 46 Define WHO and UNICEF at first use

Line 56 Provide reference for this claim

Line 57-58 Who is saying they should be? Be sure to state if this is the WHO goal

Line 62 Would be interesting if you could discuss why the increase in EBF between 2003-2008; were there certain interventions that were successful but possible excluded partners? This would be a good place to improve your justification and rational for this study

Line 62-63 Also, discuss possibilities for why EBF decreased again in 2014

Line 71 How are partners defined here? Since this is a key point of your study, should clarify who is considered a partner using the DHS data

Line 91-107 Too much discussion of the DHS methods; you can simply reference the DHS manual that explains all these details

Line 105-106 contradicts your findings. You state that 613 were identified as exclusively breastfed? So did you only include infants who were exclusively breastfed? This contradicts your coding of 0 and 1; please explain and clarify the inclusion and exclusion criteria

Line 119-129 Need rationale for the inclusion of these partner characteristics; it's concerning that a large independent variable is missing from this study; intimate partner violence and its' many forms should considered to be included; this is largely a partner characteristic that is known to influence exclusive breastfeeding; also the number of children the partner has

Line 132-137 Were antenatal visits included? If yes, please add. If not, please justify why they were not included.

Line 138-155 This can be explain by adding it to one of your results tables; does not need to be listed out in the text.

Line 157 Data Analysis: Please clarify in detail how you addressed and adjusted for the sample weights and cluster that is used in the DHS data. Also, please be specific how the binary logistic regressions were adjusted for. Describe in detail how multicollinearity was accounted for.

Line 171 This contradicts your statement that 613 children were exclusively breastfed

Table 1 Title should be more descriptive in nature; consider adding the sample sizes and n

Table 3 is very confusing. Review other PloS One articles for information on how to construct these types of tables.

Line 262-263 Your paper is specifically about exclusively breastfeeding and not maternal health; consider revising this statement to more accurately reflect your study

Line 271 Need reference after WHO

Line 275 Too vague. Consider giving specific examples on measures and policies that need to be strengthened. What types of interventions have worked well in other countries? There are examples of programs that involve partners to improve breastfeeding.https://journals.sagepub.com/doi/abs/10.1177/0890334408323545?casa_token=1F5hkpPBVpcAAAAA:ZwVCiiboCCN616qR5YLnd-miBsRO_pHq_Qg-SltAWJGnK5_rbsJVZ-H9Wr-MRqVXqneiuSjON_dJ
https://www.mdpi.com/1660-4601/17/2/413
https://www.sciencedirect.com/science/article/pii/S1871519219300654?casa_token=VGbKnwR_KQUAAAAA:zDkbBiALfqtUw2Y46kYEy46P5DNjmAM515gs2W5eTeloZ4xvw7PeBIunfD-iOtyW2_94cp5I0Q

Line 287-298 This needs to be explained more. Many studies find that higher maternal education is associated with formula feeding and non-exclusive breastfeeding. It is possible this is a statistical error based on sample sizes within the education variable or may be an error related to unaccounted for multicollinearity.

The discussion is mostly about mother's characteristics. The title and justification of the study was built on the premise of partner's characteristics. Consider revamping the study if you want to look at maternal characteristics as well. If the paper is about partner characteristics then the discussion needs to adequately reflect that. Discussion should also include specific recommendations to policy makers, public health professionals, and detailed recommendations on intervention design. What should be changed in Ghana because of the study results to improve exclusive breastfeeding in Ghana? Give examples of what has worked in other countries involving fathers and partners to improve breastfeeding. Could those be used in Ghana? What would be possible barriers? What type of future research should be done because of your study?

Double check references to make sure they are accurate

Reviewer #2: Recommendations and overarching statements –

First and foremost, I would like to congratulate the authors on their hard work. The topic and objectives of the research paper are pertinent to the current need in the maternal and child health literature. The paper is full of potential and opportunity to highlight a key research area that can guide interventions to come.

Secondly, and perhaps most importantly, the research paper in its current form requires some major modifications for the crux of the research to be brought out. Currently, the paper misses the objective it set out to achieve and needs to align with the research question again. There is an emphasis on maternal characteristics instead of paternal ones. I would strongly recommend that the authors use the Strobe checklist to present their research.

Major revisions –

1. The objective of the article is to establish and comment on the pattern between exclusive breastfeeding and partner characteristics, however, the abstract does not mention this relationship as much as it needs to. There seems to be a greater emphasis on maternal characteristics instead which defeats the purpose. The authors need to bring out the effect of paternal characteristics more.

2. The list of paternal characteristics that influence breastfeeding is certainly not exhaustive, however, some key paternal characteristics have been left out of consideration. This includes – paternal religion, paternity leave, paternal ethnicity, children fathers have had, father’s place of residence, etc. These indicators are generally available in the DHS surveys and should be available in the data dump the authors obtained. Additionally, similar factors have been considered for mothers but not for fathers who are the primary subject of this study. I believe the author’s need to revisit their overall study objective and align the methodology of the paper accordingly.

3. The entire discussion section beyond paragraphs 1 and 3 (page 22) talks only about maternal characteristics. The authors need to revisit this entire section.

4. Page 22, para 3, line 1 – The line states that “the findings indicate that partner’s occupation was the only characteristic of a partner that influenced the practice of exclusive breastfeeding”. This statement although true in the given context of the study is appreciated, would only be valid if supported with a more extensive assessment of other characteristics (mentioned in point 2)

5. Page 23, para 1, line 1 “This probably enables them to understand the need and importance to practice exclusive breastfeeding as well as motivating their partners/women to practice their exclusive breastfeeding”. This is a very broad and unsubstantiated statement and feels more like conjecture than fact. Every statement commenting beyond the results of the paper should be cited appropriately.

Minor but recommended revisions–

1. Page 22, para 1, line 3 – “relevant” to “relevance”

2. The discussion section of a paper is meant to fit the results and observations of a study in the wider context of existing literature. Some interesting parallels can be made by drawing on other aspects of maternal and child health. For instance, are similar trends seen in the case of infant malnutrition, maternal health-seeking behavior etc.

3. The authors can also add a paragraph in the introduction to explain the cultural context of breastfeeding in Ghana. Since breastfeeding often has social and cultural significance, it would help the readers understand if that might have any potential impact on overall breastfeeding practice in Ghana.

6. PLOS authors have the option to publish the peer review history of their article (what does this mean?). If published, this will include your full peer review and any attached files.

Reviewer #1: No

Reviewer #2: **Yes: **Prerna Gopal

---

## [Author Response · Author response to Decision Letter 0]

6 Apr 2021

Academic Editor 

Response: We thank the academic editor for the comment and suggestion. We have revised the entire manuscript to ensure it meets PLOS ONE requirements. 

2. We note that you have indicated that data from this study are available upon request. PLOS only allows data to be available upon request if there are legal or ethical restrictions on sharing data publicly.

Response: We apologise for the error. The Data Availability Statement should be modified to read “Data are available from the Measure Demographic and Health Survey Program (visit https://www.dhsprogram.com/data/ ) for researchers who meet the criteria for access to confidential data. The data is labeled Ghana: Standard DHS, 2014 and it can be accessed on the following link: https://dhsprogram.com/data/dataset/Ghana_Standard-DHS_2014.cfm?flag=0”.

Response: We have indicated in the revised cover letter that “The Measure Demographic and Health Survey program is authorised to distribute the 2014 Ghana Demographic and Health Survey data. If a researcher needs the data for research purpose, he/she is required to register with the Measure Demographic and Health Survey program to gain unrestricted access to the data”.

Response: The datasets used for the analysis have been attached as Supplementary Information files and we have indicated in the revised cover letter that “We have also attached the files we analysed as Supplementary Information files” (last sentence on page 1).

5. For studies involving humans categorized by race/ethnicity, age, disease/disabilities, religion, sex/gender, sexual orientation, or other socially constructed groupings, authors should: 1) Explicitly describe their methods of categorizing human populations, 2) Define categories in as much detail as the study protocol allows, 3) Justify their choices of definitions and categories, 4) Explain whether (and if so, how) they controlled for confounding variables such as socioeconomic status, nutrition, environmental exposures, or similar factors in their analysis.

Response: We have described how we categorised characteristics of respondents which are social constructs and how we controlled for confounding variables in the revised cover letter as follows: “In our study, we used characteristics of respondents such as sex, age, ethnicity, and religion. We followed the categorization of these variables used in the 2014 Ghana Demographic and Health Survey manual. We checked for multicollinearity using Variation Inflation Factor (VIF) and highly correlated variables such as maternal place of residence and religion were excluded from the analysis” (page 2, paragraph 1). 

Reviewer 1

1. Consider making the title more descriptive.

Response: We thank the reviewer for the suggestion. We have changed the wording of the topic to make it more descriptive. The revised topic is “The association between paternal characteristics and exclusive breastfeeding in Ghana” (page 1, line 1).

2. Rationale/justification for the study needs to be stronger. What specifically is this adding to the current literature

Response: We have modified the rationale/justification for the study in the revised manuscript. The study adds to the exclusive breastfeeding literature by identifying partner characteristics that significantly predict the practice of exclusive breastfeeding in Ghana by using a nationally representative sample. Please see page 4, lines 79-81, and page 5, lines 82-95.

3. If the analyses were adjusted, that needs to be explained and adjusted results should be listed as AORs instead of ORs

Response: The results were adjusted and they have been explained in the revised manuscript. The partner, mother, and infant characteristics were run together at the regression stage using the ENTER METHOD. All adjusted results have been changed from ORs to AORs. See page 2, lines 29-31, and page 11, lines 191-206.

4. Description of inclusion criteria should be more specific in the abstract methods and methods section.

Response: We thank the reviewer for the comment. All infants under 6 months old with maternal and paternal characteristics were included in the present study. This, therefore, includes infants who were either exclusively breastfed or not. We have made the inclusion criteria more specific in the abstract (page 2, lines 23-24) and methods section (page 6, lines104-106) in the revised manuscript. 

5. Results should be presented with 2 decimals; remove "95%" from methods

Response: The results section of the revised manuscript has been modified and the results are presented in two decimals where applicable (see from page 8, lines 158 - page 14, line 210). In addition, we have removed the” 95%” from the methods section and replaced it with “a significance level of 0.05” (page 8, lines 153-154).

6. Conclusion in abstract should be more specific. How do these findings help? How can they inform policies and interventions? What should public health professionals do differently in Ghana based on your findings?

Response: We have revised the conclusion in the abstract and the main manuscript to include what public health professionals can do differently. Public health professionals can adopt a couple-oriented approach to educate and counsel both mothers and partners on the importance of exclusive breastfeeding in Ghana (page 2, lines 34-36, and page 19, lines 319-320).

7. Are all authors in the same institution? If yes, that should be clarified on the title page

Response: At the time of submission of the manuscript, all authors were from the same institution. Currently, the first and third authors have moved to a different institution (the University of Environment and Sustainable Development). This has been clarified on the title page (page 1, lines 7-10).

8. Line 18 should read sub-Saharan

Response: We have modified the background of the abstract in the revised manuscript and the phrase “sub Saharan” has been deleted from it (page 1, lines 18-19). 

9. Line 43 Odd to reference EBF as an intervention; consider calling it an "essential practice"

Response: We agree with the comment of the reviewer. We have changed the word “intervention” in the introduction to “practice” in the revised manuscript (page 3, line 39). Also, we have changed “exclusive breastfeeding” to “EBF” almost throughout the entire revised manuscript.

10. Line 46 Define WHO and UNICEF at first use

Response: We have defined “WHO” and “UNICEF” in the revised manuscript (page 3, lines 42-43). 

11. Line 56 Provide reference for this claim 

Response: Thanks for the observation. We have provided Olufunlayo et al., 2019 as a reference to support our claim on page 3, line 51.

12. Line 57-58 Who is saying they should be? Be sure to state if this is the WHO goal

Response: In the revised manuscript, we have indicated who made the estimation and it now reads “The WHO estimates that at least 50% and 70% of infants aged under 6 months globally should be exclusively breastfed by 2025 and 2030 respectively” (page 3, line 52).

13. Line 62 Would be interesting if you could discuss why the increase in EBF between 2003-2008; were there certain interventions that were successful but possible excluded partners? This would be a good place to improve your justification and rationale for this study

Response: In the revised manuscript, we have included policies and programmes that may have contributed to the increase in Ghana’s EBF rate between 2003 and 2008 as follows: “The increase in Ghana’s EBF rate can be attributed to the implementation of policies and programmes such as the Breastfeeding Promotion Regulations 2000 [14], Imagine Ghana Free of Malnutrition [15], and Infant and Young Child Feeding Strategy [16]. These policies and programmes highlighted the importance of practicing exclusive breastfeeding. For instance, the Breastfeeding Promotion Regulations 2000 prohibits the sale and promotion of infant formula or any other product marketed as appropriate for feeding infants less than six months of age in health facilities or public places [14]. It is worth noting that these policies and programmes placed less emphasis on couple-oriented approaches to enhance exclusive breastfeeding” (page 5, line 55 to page 6, line 61).

14. Line 62-63 Also, discuss possibilities for why EBF decreased again in 2014

Response: We have explained plausible reasons for the decline in EBF in 2014 in the revised manuscript (page 4, lines 64-70).

15. Line 71 How are partners defined here? Since this is a key point of your study, should clarify who is considered a partner using the DHS data 

Response: We defined partners as the biological father of the child irrespective of the marital status (page 4, lines 79-80). 

16. Line 91-107 Too much discussion of the DHS methods; you can simply reference the DHS manual that explains all these details

Response: We agree with the comment of the reviewer. We have reduced the discussion of the DHS methods and have referenced the DHS manual for detailed information (page 5, lines 102-103). 

17. Line 105-106 contradicts your findings. You state that 613 were identified as exclusively breastfed? So did you only include infants who were exclusively breastfed? This contradicts your coding of 0 and 1; please explain and clarify the inclusion and exclusion criteria

Response: Thank you for the observation. The 613 included all infants under six months who were either exclusively breastfed or not. However, with the inclusion of other variables suggested by the reviewers the sample size has currently reduced to 180. This includes all infants under six months with information on both maternal and paternal characteristics. It also includes those who were exclusively breastfed as well as those who were not exclusively breastfed. See from page 5, line 103 to page 6 line 106. 

18. Line 119-129 Need rationale for the inclusion of these partner characteristics; it's concerning that a large independent variable is missing from this study; intimate partner violence and its' many forms should be considered to be included; this is largely a partner characteristic that is known to influence exclusive breastfeeding; also the number of children the partner has

Response: We acknowledged that some partner characteristics were missing in the first manuscript. This has been rectified and we have included other partners’ characteristics such as husband desire for children, religion, ethnicity, place of residence, partner’s attitude towards wife beating, and number of children by partner in the revised manuscript. See from page 6, line 118 to page 7, line 133. 

19. Line 132-137 Were antenatal visits included? If yes, please add. If not, please justify why they were not included.

Response: Antenatal visits was included in the first manuscript we submitted for review and it is also included in the revised manuscript (page 7, line 142). 

20. Line 138-155 This can be explained by adding it to one of your results tables; does not need to be listed out in the text.

Response: The section on the control variables has been summarised (page 7, lines 136-145). 

21. Line 157 Data Analysis: Please clarify in detail how you addressed and adjusted for the sample weights and cluster that is used in the DHS data. Also, please be specific how the binary logistic regressions were adjusted for. Describe in detail how multicollinearity was accounted for.

Response: The data were weighted using the sampling weight variable (v005). A weight variable was generated by dividing the variable (v005) by 1,000,000. This was applied throughout the various stages of the analysis. The binary logistic regression was adjusted by using the enter mode in the mode where all variables were put in the model and run at the same time. In addition, we checked for multicollinearity and highly correlated variables such as mothers' place of residence and religion were excluded from the analysis because they were collineated. 

22. Line 171 This contradicts your statement that 613 children were exclusively breastfed

Response: The entire results section of the revised manuscript has been modified (page 8, lines 158 - page 14, line 210).

23. Table 3 is very confusing. Review other PloS One articles for information on how to construct these types of tables

Response: We thank the reviewer for the observation. We have reviewed other PloS One articles and revised all tables in the revised manuscript. 

24. Line 262-263 Your paper is specifically about exclusively breastfeeding and not maternal health; consider revising this statement to more accurately reflect your study

Response: We have modified the sentence to accurately reflect exclusive breastfeeding. We changed the phrase “maternal health” to “EBF” in the revised manuscript (page 14, line 215). 

25. Line 271 Need reference after WHO

Response: We thank the reviewer for the observation. The statement has been referenced in the revised manuscript (page 15, line 223).

26. Line 275 Too vague. Consider giving specific examples on measures and policies that need to be strengthened. What types of interventions have worked well in other countries? There are examples of programs that involve partners to improve breastfeeding

Response: We have included specific programmes in Italy, Brazil and Taiwan, and their impact on both breastfeeding and exclusive breastfeeding in the revised manuscript (page 15, line 231-237).

27. Line 287-298 This needs to be explained more. Many studies find that higher maternal education is associated with formula feeding and non-exclusive breastfeeding. It is possible this is a statistical error based on sample sizes within the education variable or may be an error related to unaccounted for multicollinearity.

Response: We have revised the manuscript to account for multicollinearity. Variables such as maternal place of residence and religion were collineated and consequently, they were excluded from the analysis. 

28. The discussion is mostly about mother's characteristics. The title and justification of the study was built on the premise of partner's characteristics. Consider revamping the study if you want to look at maternal characteristics as well. If the paper is about partner characteristics then the discussion needs to adequately reflect that. Discussion should also include specific recommendations to policy makers, public health professionals, and detailed recommendations on intervention design. What should be changed in Ghana because of the study results to improve exclusive breastfeeding in Ghana? Give examples of what has worked in other countries involving fathers and partners to improve breastfeeding. Could those be used in Ghana? What would be possible barriers? What type of future research should be done because of your study? 

Response: We have included more other partner characteristics such as husband desire for children, religion, ethnicity, place of residence, partner attitude towards wife beating, and number of children by partner in the revised manuscript (page 6, lines 118-120). In addition, the discussion now reflects partner characteristics and recommendations on policy to health workers. Specific programmes involving men and women that have worked in other countries have been included in the discussion. In addition, specific programmes that could be also be adopted in Ghana have been added to the revised manuscripts. Also, future research has been indicated in the manuscript. See from page 14, line 213 to page 19, line 323.

29. Double check references to make sure they are accurate

Response: We have double check references to make sure they are accurate. 

Reviewer 2

1. The objective of the article is to establish and comment on the pattern between exclusive breastfeeding and partner characteristics, however, the abstract does not mention this relationship as much as it needs to. There seems to be a greater emphasis on maternal characteristics instead which defeats the purpose. The authors need to bring out the effect of paternal characteristics more

Response: The abstract has been revised to include the influence of partner characteristics on exclusive breastfeeding (page 2, lines 18-36). 

2. The list of paternal characteristics that influence breastfeeding is certainly not exhaustive, however, some key paternal characteristics have been left out of consideration. This includes – paternal religion, paternity leave, paternal ethnicity, children fathers have had, father’s place of residence, etc. These indicators are generally available in the DHS surveys and should be available in the data dump the authors obtained. Additionally, similar factors have been considered for mothers but not for fathers who are the primary subject of this study. I believe the author’s need to revisit their overall study objective and align the methodology of the paper accordingly.

Response: We thank the reviewer for the observation: We have included other partner characteristics such as husband desire for children, religion, ethnicity, place of residence, partner attitude towards wife beating, and number of children by partner in the revised manuscript (page 6, lines 118-120).

3. The entire discussion section beyond paragraphs 1 and 3 (page 22) talks only about maternal characteristics. The authors need to revisit this entire section. 

Response: The entire discussion section has been modified to reflect the influence of partner characteristics and exclusive breastfeeding in the revised manuscript. See from page 14, line 213 to page 19, line 311.

4. Page 22, para 3, line 1 – The line states that “the findings indicate that partner’s occupation was the only characteristic of a partner that influenced the practice of exclusive breastfeeding”. This statement although true in the given context of the study is appreciated, would only be valid if supported with a more extensive assessment of other characteristics (mentioned in point 2).

Response: We agree with the reviewer’s comment. As a way of making the results robust, we have added other partner characteristics such as husband desire for children, religion, ethnicity, place of residence, partner attitude towards wife beating, and number of children by partner in the revised manuscript (page 6, lines 118-120).

5. The discussion section of a paper is meant to fit the results and observations of a study in the wider context of existing literature. Some interesting parallels can be made by drawing on other aspects of maternal and child health. For instance, are similar trends seen in the case of infant malnutrition, maternal health-seeking behavior etc. 

Response: We thank the reviewer for the comment. We have highlighted in the revised manuscript that although Ghana’s EBF rate has declined, there has been improvement in other child health indicators such as stunting, wasting, and underweight (page 15, lines 226-229).

6. The authors can also add a paragraph in the introduction to explain the cultural context of breastfeeding in Ghana. Since breastfeeding often has social and cultural significance, it would help the readers understand if that might have any potential impact on overall breastfeeding practice in Ghana.

Response: We have added a paragraph in the introduction section which talks about the influence of interpersonal and community factors on exclusive breastfeeding in Ghana in the revised manuscript (page 4, lines 71-78).

---

## [Decision Letter · Decision Letter 1]

9 May 2021

PONE-D-20-36508R1

The association between paternal characteristics and exclusive breastfeeding in Ghana

PLOS ONE

Dear Dr. Kyei-Arthur,

Thank you for submitting your manuscript to PLOS ONE. After careful consideration, we feel that it has merit but does not fully meet PLOS ONE’s publication criteria as it currently stands. Therefore, we invite you to submit a revised version of the manuscript that addresses the points raised during the review process.

Important comments have again been offered in this second peer-review that should be addressed.  In particular, please consider how empirical evidence and authors' speculation are worded throughout the Discussion.  Both can be communicated (and are valuable given the unique insight in terms of knowledge of the local context), but it is important to clearly distinguish for the reader the difference.      

We look forward to receiving your revised manuscript.

Kind regards,

Joann M. McDermid, MSc, PhD, RDN, FAND

Academic Editor

PLOS ONE

Journal Requirements:

Reviewers' comments:

Reviewer's Responses to Questions

**Comments to the Author**

1. If the authors have adequately addressed your comments raised in a previous round of review and you feel that this manuscript is now acceptable for publication, you may indicate that here to bypass the “Comments to the Author” section, enter your conflict of interest statement in the “Confidential to Editor” section, and submit your "Accept" recommendation.

Reviewer #1: All comments have been addressed

Reviewer #2: (No Response)

2. Is the manuscript technically sound, and do the data support the conclusions?

Reviewer #1: Yes

Reviewer #2: Partly

3. Has the statistical analysis been performed appropriately and rigorously? 

Reviewer #1: Yes

Reviewer #2: No

4. Have the authors made all data underlying the findings in their manuscript fully available?

Reviewer #1: Yes

Reviewer #2: Yes

5. Is the manuscript presented in an intelligible fashion and written in standard English?

Reviewer #1: Yes

Reviewer #2: Yes

6. Review Comments to the Author

Reviewer #1: The authors have thoroughly addressed each question and concern that was raised during the initial review process. I have no further questions or concerns with this manuscript.

Reviewer #2: Line no. 22 – Correction to “This cross-sectional study used data from the 2014 Ghana Demographic and Health Survey”

Line no. 30 – Replace “desire” with “desired”

Line 78-79 – Partner is defined according to empirical evidence. However, it must also be mentioned if it aligns with the definition as per the DHS data

Authors also need to recheck the language and grammar of the overall article during this stage of revision.

Methodology –

1. Was the sample representative of the ethnic, religious, occupational standards for Ghana? In either case, this should be mentioned in the methodology and then inference accordingly made in the discussion.

Discussion –

The discussion section is steeping in conjecture and needs to be empirical in nature. Every statement should either be derived from the findings of the study or should be cited accurately. The authors also need to revisit the objective of the study and align the content with it.

1. Line 231- 232 – Why is neonatal mortality being mentioned here? It hasn’t been talked about any where in the passage. How is this adding to the overall discussion?

2. Line 241- 246 – The authors need to establish who the primary focus of the study is – partners or mothers? If the objective is to compare the two, that needs to be brought out better

3. Line 251 – 252 – Is there any evidence to support this statement?

4. Line 255 – 262 - Is there any evidence to support these statements? Was the survey done during Ramadan? The relevancy of these statement is not clear or justified.

5. Line 266 – Which finding do the authors mean when they say “this finding”

6. Line 271 – 287 – How does this fit into the objective of the study – understanding partner characteristics that impact EBF?

7. Of the ~40 AOR calculated, majority are insignificant. The authors should include a segment discussing these observations.

7. PLOS authors have the option to publish the peer review history of their article (what does this mean?). If published, this will include your full peer review and any attached files.

Reviewer #1: No

Reviewer #2: No

---

## [Author Response · Author response to Decision Letter 1]

13 May 2021

Responses to the academic editor and reviewers’ comments

Academic Editor

Response: We have gone through all the references to ensure that they are complete and correct. We have removed two references (Asare et al. [38]; Sika-Bright and Oduro [46]) from the revised manuscript since they talked about the effect of mothers' education on exclusive breastfeeding. We have included new references (Ludvigsson and Ludvigsson [34]; Chudasama, Patel, and Kavishwar, [46]) in the revised manuscript to corroborate our finding that partners with low education are less likely to practice exclusive breastfeeding compared to those with no education. Also, we have modified the sentence “Therefore, formal education by itself may not necessarily empower a mother to practice EBF.” to read “Therefore, mothers whose partners have acquired some level of formal education may not necessarily be empowered to practice EBF.” Please see Line no: 252-253.

In addition, we have removed the reference Bhatta [48] from our revised manuscript since it did not support our argument in the manuscript. Also, we have deleted the reference Adsera [50] from the revised manuscript since we have deleted the argument it supports from the revised manuscript.

38. Asare BY-A, Preko JV, Baafi D, Dwumfour-Asare B. Breastfeeding practices and determinants of exclusive breastfeeding in a crosssectional study at a child welfare clinic in Tema Manhean, Ghana. International Breastfeeding Journal (2018) 13:12 2018;13(1):12.

46. Sika-Bright S, Oduro GY. Exclusive breastfeeding practices of mothers in Duakor, a traditional migrant community in Cape Coast, Ghana. Journal of Global Initiatives: Policy, Pedagogy, Perspective. 2013;8(1):87-102.

34. Ludvigsson JF, Ludvigsson J. Socio‐economic determinants, maternal smoking and coffee consumption, and exclusive breastfeeding in 10 205 children. Acta Paediatrica. 2005;94(9):1310-9.

46. Chudasama RK, Patel PC, Kavishwar AB. Determinants of exclusive breastfeeding in south Gujarat region of India. Journal of Clinical Medicine Research. 2009;1(2):102.

48. Bhatta DN. Involvement of males in antenatal care, birth preparedness, exclusive breast feeding and immunizations for children in Kathmandu, Nepal. BMC Pregnancy and Childbirth. 2013;13(1):1-7.

50. Adsera A. Differences in desired and actual fertility: An economic analysis of the Spanish case. Bonn: IZA; 2005.

Reviewer 2 

1. Line no. 22 – Correction to “This cross-sectional study used data from the 2014 Ghana Demographic and Health Survey”. 

Response: We thank the reviewer for the comment and observation. This has been corrected in the revised manuscript. Please see Line no. 22.

2. Line no. 30 – Replace “desire” with “desired”

Response: The word “desire” has been changed to “desired” in the revised manuscript. Please see Line no. 30.

3. Line 78-79 – Partner is defined according to empirical evidence. However, it must also be mentioned if it aligns with the definition as per the DHS data

Response: We thank the reviewer for the comment. We have revised the definition of a partner in the revised manuscript to align with the DHS definition. A partner is defined as “a man who is either currently married or living with a woman”. Please see Line no. 79. 

4. Was the sample representative of the ethnic, religious, occupational standards for Ghana? In either case, this should be mentioned in the methodology and then inference accordingly made in the discussion.

Response: Ethnicity, religion, and occupation are aligned with the standards for Ghana. We have included the statement “All variables and their categories used in this study align with the standards for Ghana.” in the revised manuscript to that effect. Please see Line no. 145-146. 

5. Line 231- 232 – Why is neonatal mortality being mentioned here? It hasn’t been talked about any where in the passage. How is this adding to the overall discussion?

Response: We thank the reviewer for the comment. Neonatal mortality is very important with regard to exclusive breastfeeding. In that non-exclusive breastfeeding has a consequence on mortality. We have therefore modified the sentence “It reduces infant mortality and morbidity, enables infants to crawl early, enhances growth and cognitive development, and reduces the risk of childhood obesity [3-7].” to read “It reduces neonatal and infant mortality and morbidity, enables infants to crawl early, enhances growth and cognitive development, and reduces the risk of childhood obesity [3-7].” in the introduction section to that effect. Please see Line no. 46-47.

6. Line 241- 246 – The authors need to establish who the primary focus of the study is – partners or mothers? If the objective is to compare the two, that needs to be brought out better

Response: We thank the reviewer for the observation and comment. Partners are the primary focus of this study. We used mothers’ education to explain partners’ education. We have therefore modified it in the revised manuscript. Please see Line no. 247-249. In addition, we have modified the sentence “It was also evident in the findings that the likelihood to exclusively breastfeed was lower among Muslim mothers compared to Christian mothers.” to read “It was also evident in the findings that the likelihood to exclusively breastfeed was lower among mothers whose partners were Muslims compared to those whose partners were Christians.” Please see Line no. 259-260.

7. Line 251 – 252 – Is there any evidence to support this statement?

Response: The statement “Therefore, partners who desire more children may discourage the practice of EBF to ensure their fertility desire is accomplished.” is speculative and we have deleted it from the revised manuscript. 

8. Line 255 – 262 - Is there any evidence to support these statements? Was the survey done during Ramadan? The relevancy of these statement is not clear or justified.

Response: We thank the reviewer for the comment. We have modified the paragraph in question as follows: “It was also evident in the findings that the likelihood to exclusively breastfeed was lower among mothers whose partners were Muslims compared to those whose partners were Christians. This finding is consistent with previous studies [23]. The Holy Quran encourages breastfeeding among Muslims [48]. However, the reasons why mothers whose partners were Muslims are less likely to exclusively breastfeed compared to those whose partners are Christians could not be explained in the present study and therefore further studies using qualitative approaches could explore the probable reasons.” Please see Lines no. 259-265.

9. Line 266 – Which finding do the authors mean when they say “this finding”

Response: We thank the reviewer for the comment. We have revised the manuscript and deleted the section “This finding however contradicts our finding that mothers whose partners desire more children were less likely to practice EBF compared to those whose partners desire fewer children. A probable reason for the discrepancy in the findings is that there is a difference between actual and desired fertility. Desired fertility may not necessarily translate into actual fertility due to several factors including labour market and economic conditions [49].”

10. Line 271 – 287 – How does this fit into the objective of the study – understanding partner characteristics that impact EBF?

Response: We thank the reviewer for the comment. Although our study focused on the influence of partner characteristics on exclusive breastfeeding, mothers and infant characteristics also influence exclusive breastfeeding. Consequently, we controlled for these characteristics. Since the mother’s antenatal care visits and age of infant were found to be significantly associated with exclusive breastfeeding in this study, we, therefore, discussed it in the “Discussion Section”.

11. Of the ~40 AOR calculated, majority are insignificant. The authors should include a segment discussing these observations.

Response: We thank the reviewer for the suggestion. We have discussed the non-significant paternal characteristics in the “Discussion Section”. Please see Line no. 269-278.

---

## [Editor Report · Decision Letter 2]

18 May 2021

The association between paternal characteristics and exclusive breastfeeding in Ghana

PONE-D-20-36508R2

Dear Dr. Kyei-Arthur,

We’re pleased to inform you that your manuscript has been judged scientifically suitable for publication and will be formally accepted for publication once it meets all outstanding technical requirements.

Kind regards,

Joann M. McDermid, MSc, PhD, RDN, FAND

Academic Editor

PLOS ONE
---

## [Editor Report · Acceptance letter]

24 May 2021

PONE-D-20-36508R2 

The association between paternal characteristics and exclusive breastfeeding in Ghana 

Dear Dr. Kyei-Arthur:

I'm pleased to inform you that your manuscript has been deemed suitable for publication in PLOS ONE. Congratulations! Your manuscript is now with our production department. 

Kind regards, 

on behalf of

Professor Joann M. McDermid 

Academic Editor

PLOS ONE